# Identification of the reporter gene combination that shows high contrast for cellular level MRI

**Naoya Hayashi[1,2,3], Junichi Hata[1,2,4] \*, Tetsu Yoshida[2,4], Daisuke Yoshimaru[2,5], Yawara Haga[2], Hinako Oshiro[1,2], Ayano Oku[1,2], Noriyuki Kishi[2,4], Takako Shirakawa[1], Hideyuki Okano[2,4]**

**1** Graduate School of Human Health Sciences, Tokyo Metropolitan University, Tokyo, Japan, **2** RIKEN, Center for Brain Science, Wako, Saitama, Japan, **3** Department of Radiology, Tokyo Medical University Hospital, Tokyo, Japan, **4** Graduate School of Medicine, Keio University, Tokyo, Japan, **5** Division of Regenerative Medicine, The Jikei University School of Medicine, Tokyo, Japan

\* j-hata@tmu.ac.jp

**Data Availability Statement:** All MRI images and Western blotting images are available via the Open Science Framework at: Direct URL: https://osf.io/45uax/?view_only=

## Abstract

Currently, we can label the certain cells by transducing specific genes, called reporter genes, and distinguish them from other cells. For example, fluorescent protein such as green fluorescence protein (GFP) is commonly used for cell labeling. However, fluorescent protein is difficult to observe in living animals. We can observe the reporter signals of the luciferin-luciferase system from the outside of living animals using *in vivo* imaging systems, although the resolution of this system is low. Therefore, in this study, we examined the reporter genes, which allowed the MRI-mediated observation of labeled cells in living animals. As a preliminary stage of animal study, we transduced some groups of plasmids that coded the protein that could take and store metal ions to the cell culture, added metal ions solutions, and measured their T1 or T2 relaxation values. Finally, we specified the best reporter gene combination for MRI, which was the combination of transferrin receptor, DMT1, and Ferritin-M6A for T1WI, and Ferritin-M6A for T2WI.

## Introduction

With recent advancements, we can label the certain cells using gene transfer techniques and distinguish them from other cells. This labeling technology has made cell movement or mechanism tracking possible. The genes used for this technology are called reporter genes [1], such as fluorescent proteins. One example of fluorescent protein is green fluorescent protein (GFP), which emits green light under blue light excitation, and the cells expressing GFP are thus labeled by green fluorescence [2]. GFP is not harmful to living animals and does not require any enzymatic reactions to function. Therefore, GFP is frequently used in the fields of medical and biological sciences. However, GFP also has disadvantages, such as GFP expressed in-depth in living animals cannot be observed from the outside. Therefore, GFP expression should be observed on sliced specimens using microscopes. The GFP can be observed from the outside

345a1e5fa40745468c4a820e0be8d9df DOI: 10.17605/OSF.IO/45UAX.

**Funding:** H.O. JP15dm0207001 and JP22bm0304003 Japan Agency for Medical Research and Development (AMED) https://www.amed.go.jp/ The funders had no role in study design, data collection and analysis, decision to publish, or preparation of the manuscript. J.H. JP20H03630 Japan Society for the Promotion of Science (JSPS) KAKENHI https://www.jsps.go.jp/j-grantsinaid/ The funders had no role in study design, data collection and analysis, decision to publish, or preparation of the manuscript. J.H. JPMXS0450400622 "MRI platform" as a program of the Project for Promoting Public Utilization of Advanced Research Infrastructure of the Ministry of Education, Culture, Sports, Science and Technology MEXT, Japan https://mripf.jp/ The funders had no role in study design, data collection and analysis, decision to publish, or preparation of the manuscript.

**Competing interests:** The authors have declared that no competing interests exist.

in nude mice, and cannot be applied to bigger animals like macaques or marmosets [3]. As another example, luciferin–luciferase reaction offers an alternative cell labeling method [4]. The luminous phenomenon from this reaction can be captured by a machine called *in vivo* imaging system (IVIS) even from the depth of living animals [5]. However, the resolution of the IVIS images is very low, it is thus impossible to observe luciferin–luciferase reaction at the microscopic level. Therefore, reporter genes are required which can be observed in-depth in living animals and at high resolution.

Meanwhile, Magnetic Resonance Imaging (MRI) related reporter genes, such as transferrin receptor, Divalent Metal Transporter 1 (DMT1), Ferritin, Ferritin-M6A, and Mms6, are gathering attention. The transferrin receptor allows intracellular transferrin transport [6, 7]. Most iron in animals is connected to transferrin, and transferrin receptor thus substantial role a role in taking iron into the cells. The transferrin receptor can reportedly also take other metal like Mn ions [6]. The DMT1 receives the metal ions from the transferrin receptor, and transports metal ions to their appropriate destinations in the cell [8, 9]. Ferritin receives Fe ions from the metal transporter like DMT1 and stores them [10, 11]. Mms6 is a protein in magnetic bacteria with the role of maintaining the iron-joined structure called magnetosome [12]. Ferritin-M6A is an artificial protein made by the fusion between Ferritin and a part of Mms6 by a peptide bond [13]. Ferritin-M6A is reportedly able to store more iron in the cell compared with Ferritin.

All the five aforementioned proteins are reportedly useful reporters for MRI [6–12]. However, no study has elucidated which one is the best reporter yet. In addition, it has not been revealed whether a stronger contrast on MRI could be obtained upon the combination of these reporter genes. Therefore, we transferred these five reporter genes into the cell culture and compared them. We also co-expressed two or three genes in the cells and examined which combination or individual reporter gene could be the most suitable.

## Material and methods

### Cell culture and transfection

We used Human Embryonic Kidney 293T (HEK293T) cells (from the American Type Culture Collection, Manassas, VA, USA) for culture. Fig 1A shows the microscopic images of HEK293T cells. The microscope information is as follows: BZ-X710(KEYENCE, Osaka, Japan), magnification, x10; phase contrast objective lens, BZ-PF10P (KEYENCE); fluorescent filter for GFP, OP-87762(excitation wavelength = 360/40 nm, absorption wavelength = 460/50 nm, dichroic mirror wavelength = 400 nm, KEYENCE). The HEK293T cells are reported to be suitable for gene transfer because of their expression of SV40 Large T antigen [14, 15]. We used Dulbecco's modified eagle medium (08459–64, Nacalai Tesque, Kyoto, Japan) as culture medium with 10% fetal bovine serum and 1% penicillin streptomycin (P/S). We used the cells with a maximum passage number of 20. We passaged $6 \times 10^5$ HEK293T cells to 6-well plates with 2 mL culture media per well and incubated at 37˚C in a 5% $CO_2$ incubator. On the subsequent day, we transfected the cells with the reporter genes by lipofection, using the commercial Lipofectamine 3000 Reagent kit (Thermo Fisher Scientific, Waltham, MA, USA). The plasmids for transfection were as follows: CAG-FLAG-transferrin receptor, CAG-FLAG-DMT1, CAG-FLAG-Ferritin, CAG-FLAG-Ferritin-M6A, CAG-FLAG-Mms6, and CAG-Venus (Fig 1B); all were synthetic entry plasmids. They were synthetic plasmids that we requested FASMAC (Kanagawa, Japan) to create. We transfected the cells with these plasmids in the following combinations: (A): Transferrin receptor + DMT1 + Ferritin; (B): Transferrin receptor + DMT1 + Ferritin-M6A; (C): Transferrin receptor + DMT1 + Mms6; (D): Transferrin receptor + DMT1 + Venus; (E): Ferritin-M6A + Venus; (F): Venus (Control). Table 1 shows the amount of each plasmid. To prevent the plasmid amount-related bias, identical gene amounts

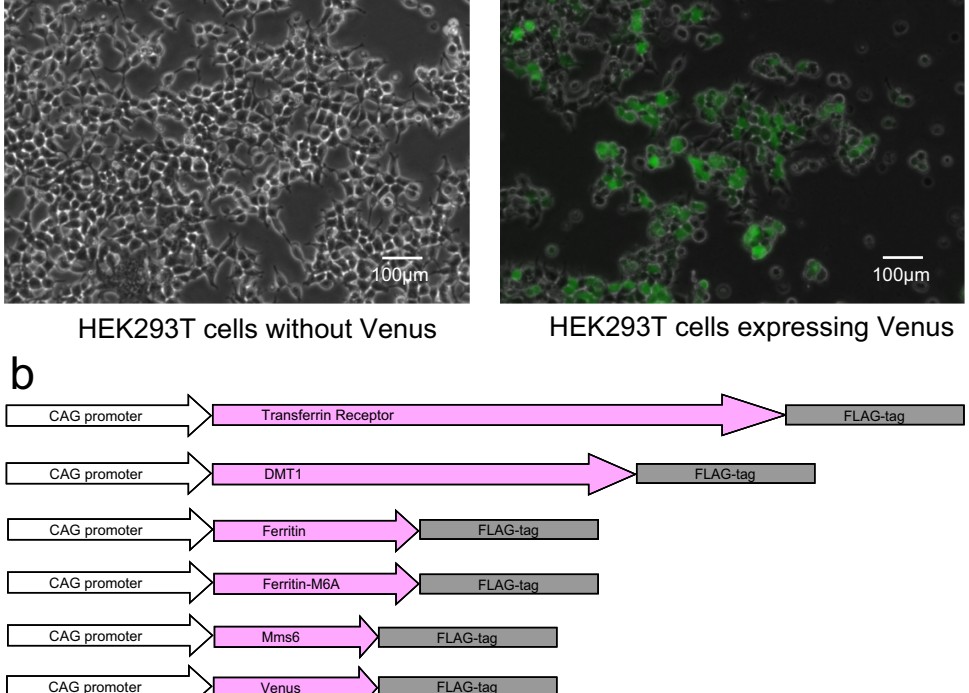

**Fig 1. Microscopic images of HEK293T cells and the constructs of the plasmids.** Left: microscopic image of HEK293T cell in the bright field mode; Right: microscopic image of HEK293T cells expressing GFP, overlaying bright field and the GFP channel (a). Microscopic observation of GFP expression in the cells confirmed that they were not strange cells and were not dead. All the plasmids we transfected to the cells had CAG promoter (b). In addition, the plasmids associated with the metal ions had a FLAG tag for Western blotting.

were maintained. The total transgenic plasmid amount was adjusted with Venus, the control. We transfected each of these combinations four times, each of which was measured using MRI. Therefore, the biological replicates of this study were four.

## Western blotting

We performed Western blotting to verify the expressed protein levels in the cells upon transfection. Western blot is a molecular biology method to verify the presence of specific proteins [16, 17]. We collected the cells expressing proteins with the Laemmli sample buffer (#1610747,

**Table 1. Reporter gene quantities for each transfection.**

|  | TfR | DMT1 | Fn | Fn-M6A | Mms6 | Venus |
|---|---|---|---|---|---|---|
| (A) | 0.83 μg | 0.83 μg | 0.83 μg | - | - | - |
| (B) | 0.83 μg | 0.83 μg | - | 0.83 μg | - | - |
| (C) | 0.83 μg | 0.83 μg | - | - | 0.83 μg | - |
| (D) | 0.83 μg | 0.83 μg | - | - | - | 0.83 μg |
| (E) | - | - | - | 0.83 μg | - | 1.67 μg |
| (F) | - | - | - | - | - | 2.5 μg |

Each reporter gene quantity for MRI was maintained at the same amount (0.83 μg) to avoid bias of the plasmid quantity. For lipofection with the gene transfer method applied in this experiment, the total amount of plasmids had to be 2.5 μg, thus we supplemented the shortage with Venus.

BIO-RAD, Hercules, CA, USA) including mercaptoethanol and boiled them at 100˚C to break the disulfide bonds. Subsequently, we separated the proteins by performing electrophoresis (20 mA, 40 min) using 10% Mini-PROTEAN TGX Gels (BIO-RAD, Hercules, CA, USA). The proteins on the gel were transferred to membranes using the Trans-Blot Turbo Transfer System (BIO-RAD). To block the no protein area of the poly vinyl difluoride (PVDF) membranes, they were exposed to 5% skimmed milk dissolved in Tris Buffered Saline with Triton X (TBS-T), shaken for 30 min, and washed with TBS-T. After washing, the membranes were incubated with anti-FLAG antibody (Anti-DDDDK tag; rabbit, F7425, Merck, Darmstadt, Germany) at 4˚C overnight. After washing with TBS-T, the membranes were incubated with antirabbit IgG (HRP) antibody (ab6721, Abcam Inc., Cambridge, CB2 0AX, UK) and shaken for 2 h. After washing with TBS-T, we stained the protein-bound antibodies with Clarity Western ECL Substrate (BIO-RAD, Hercules, CA, USA), and observed with a Lumino imaging Analyzer (LAS-3000, FUJIFILM, Tokyo, Japan).

## Exposure to metal ion and MRI scanning

We supplemented the cultures with metal ion solution 2 days after the transfection. The metal ion solution comprised a 10 μL Holo-transferrin (10 mg/mL) or 50 μL $MnCl_2$ (200 mg/mL) solution. We exposed the cells to the metal ion solution for 3 h. After the exposure, we removed the culture by centrifugation (3000 round per minute (rpm), 1 min), and washed the cells with phosphate buffered saline (PBS) to clear the metal ions from the surface of the cells. The washed cells were moved into 0.2 mL tubes with a pointed tip and the tubes were filled with PBS. The 0.2 mL tubes were fixed into a dedicated holder, and the holder was fixed into a 25 mL centrifuge tube (Fig 2A). We placed the cells in the centrifuge tube into the 28 mm

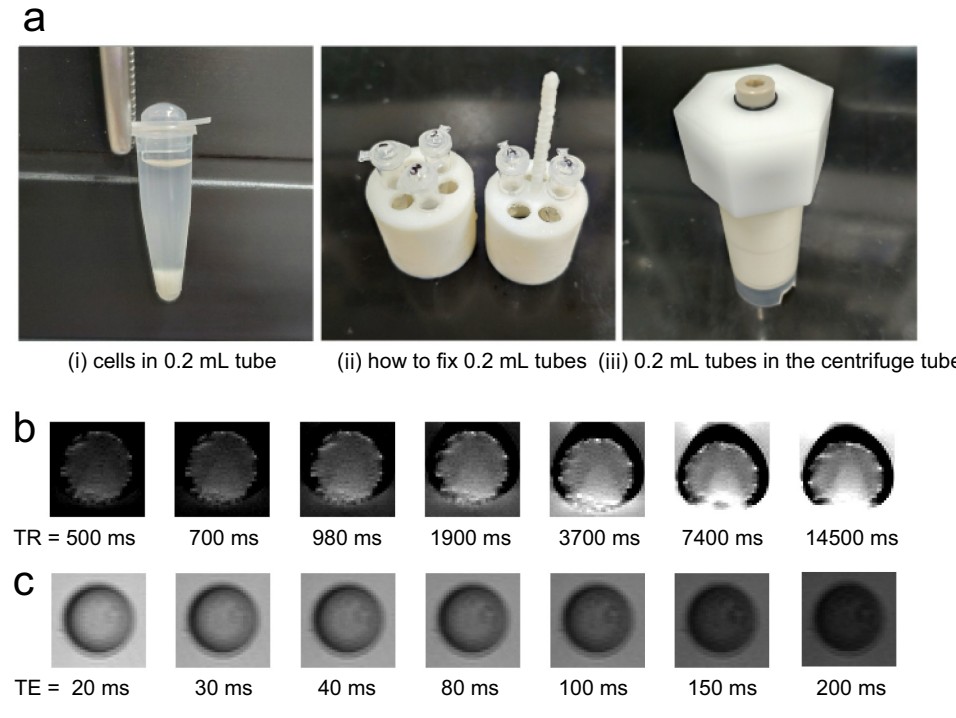

**Fig 2. Cell fixation method for MRI, acquired raw MR images.** The cells in tubes were fixed as follows: after exposure to $MnCl_2$ or Holo-transferrin, the cells were washed with PBS and collected in 0.2 mL tubes with a pointed tip with PBS (a-i). The 0.2 mL tubes were subsequently fixed in a dedicated holder (a-ii) and placed in a 25 ml centrifuge tube for MRI imaging (a-iii). Fig 2B and 2C show the parts of the acquired MR images.

solenoid system coil (Takashima Seisakusho, Yokohama, Japan) and measured the T1 or T2 relaxation values. We used a 9.4 T ultrahigh field MRI (Bruker, Biospin94/30, Max Gradient Strength 660 mT/m, Ettlingen, Germany) in the RIKEN Center for Brain Science. The MRI scanning conditions were as follows. T1 relaxation value measurement: rapid acquisition with relaxation enhancement; repetition time (TR), 500–20247 ms (12 steps); echo time (TE), 9.8 ms; resolution, 500 × 500 um; slice thickness, 1.0 mm; and average, 1. T2 relaxation value measurement: multislice multiecho (MSME); TR, 6000 ms; TE, 10–1500 ms (150 steps); resolution, 500 × 500 um; slice thickness, 1.0 mm; and averages, 1. We calculated the T1 or T2 relaxation values and created T1 or T2 maps by using MATLAB 2020b (MathWorks, Natick, MA, USA). The statistical significance test was performed using analysis of variance and Bonferroni methods. The significance level was set at $p = 0.05$.

## Results

### Constructs of the plasmid and transfection

Fig 1B shows the constructs of the plasmid we transfected in this study. All plasmids had the CAG-promoter and FLAG tag. CAG is one of the promoters for forced expression in all cells [18]. FLAG is the tag for Western blot, a polypeptide comprising 8 amino acids. FLAG tag is known to not affect protein function if fused to the appropriate position on the appropriate protein [19]. We used Venus as control and to confirm the successful gene expression. Venus is a modified GFP version with a stronger emission than GFP. The 48th amino acid of GFP is phenylalanine, which is changed into leucine in Venus.

### Western blotting

Fig 3 delineates the results of the Western blotting. The bands corresponding to all the reporter genes we used in this study were on the appropriate site. Therefore, we proved that the plasmids we used in our experiments functioned properly and that the transfection was successful, whether alone or in combination.

### T1 and T2 relaxation value measurements by MRI scanning

Fig 2B and 2C show parts of the obtained MR images. The signal intensity at each TR or TE was measured from the images and approximated by an exponential function (Figs 4A and 5A). From the approximation, T1 or T2 relaxation values were calculated (Figs 4B and 5B). When the cells were exposed to $MnCl_2$, those expressing Ferritin-M6A had the lowest T1 relaxation value among Ferritin, Ferritin-M6A, and Mms6 (Fig 4B). Subsequently, we investigated whether Ferritin-M6A should be combined with transferrin receptor and DMT1. The cells expressing the transferrin receptor, DMT1, and Ferritin-M6A simultaneously showed the lowest T1 relaxation value (Fig 4B). Finally, the T1 relaxation value of the cells expressing transferrin receptor, DMT1, and Ferritin-M6A was 61% that of the control (Fig 4B).

When the cells were exposed to Holo-transferrin, the cells expressing Ferritin-M6A also had the lowest T2 relaxation value among Ferritin, Ferritin-M6A, and Mms6 (Fig 5B). In the comparison between Ferritin only and the combination of transferrin receptor, DMT1, and Ferritin-M6A, the cells expressing only Ferritin-M6A had the lowest T2 relaxation value (Fig 5B). Finally, the T2 relaxation value of the cells expressing only Ferritin-M6A was 69% that of the control (Fig 5B).

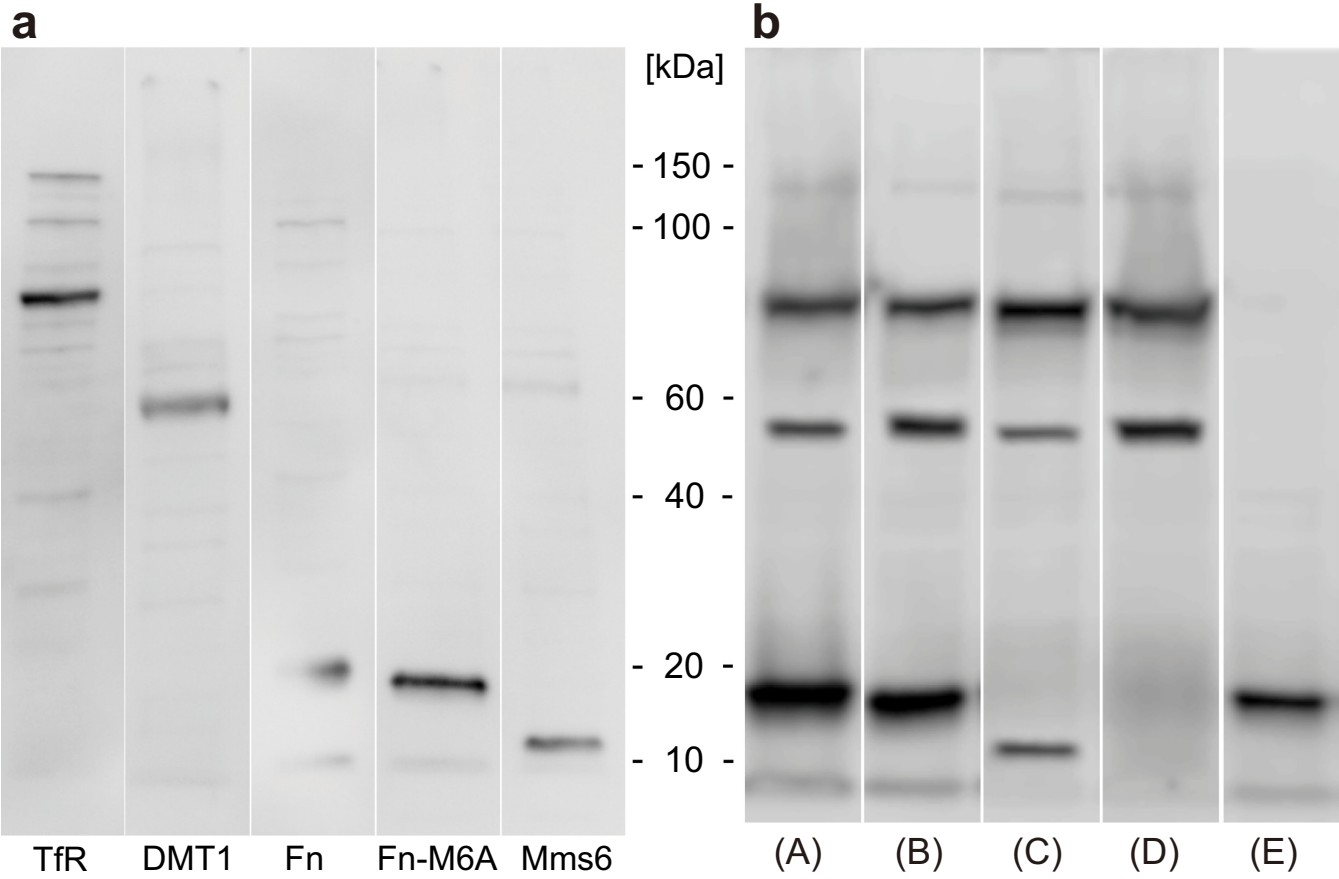

**Fig 3. Western blotting results.** The molecular weights of the proteins expressed by the reporter genes are as follows: Transferrin receptor: 84.3 kDa, DMT1: 62.3 kDa, Ferritin: 18.3 kDa, Ferritin-M6A: 18.3 kDa, and Mms6: 14.0 kDa. The results when expressed alone (a) and the results when expressed in combination (b).

## Discussion

In this study, we revealed that the combination of transferrin receptor, DMT1, and Ferritin-M6A for $MnCl_2$ and only Ferritin-M6A for Holo-transferrin were the best reporter gene combinations for MRI when $MnCl_2$ or Holo-transferrin was used as contrast media. The novelty of this study is the investigation of the optimal combination of multiple genes when expressed simultaneously. In addition, we clarified the extent of the T1 or T2 shortening effect by the specified reporter genes regarding cell culture.

First, we will discuss the validity of the T1 and T2 relaxation values measured in this study. Herein we consider the signal-to-noise ratio (SNR) and linear approximation accuracy. Regarding SNR, the MR images need an SNR of over 50 for clear object visualization [20]. Therefore, in this study, we set the scan conditions for an SNR over 60 (T1 relaxation value measurement, images of repetition time = 500 ms; T2 relaxation value measurement, images of echo time = 1000 ms). Therefore, we proved that the SNR of the acquired images was high enough to calculate the T1 or T2 relaxation values. Second, we will discuss the linear approximation accuracy. In our study, we used the linear approximation in one-logarithmic graphs to calculate the T1 and T2 relaxation values. The coefficient of determination ($R^2$) for the linear approximation was higher than 0.6 for all T1 and T2 relaxation value calculations. The coefficient of correlation of 0.7 is said to be a strong value. The $R^2$ of our study exceeded the square

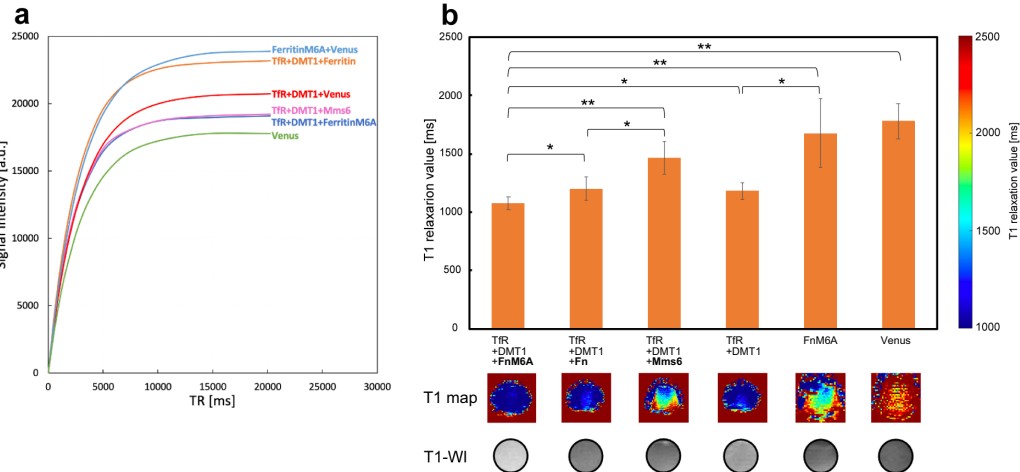

**Fig 4. T1 relaxation values of the cells expressing each gene combination when the cells were exposed to MnCl₂.** We performed MR imaging of the cells exposed to $MnCl_2$ after gene expression while changing the TR (TR = 500–20247 ms; 12 steps) (a). The T1 relaxation values are shown in the graph (*: $p < 0.05$, **: $p < 0.01$, error bar: standard deviation), and the T1 maps and T1-weighted images are shown below (b). The T1 relaxation value of the identified optimal gene group (transferrin receptor + DMT1 + Ferritin-M6A) was 61% that of the control (Venus).

of 0.7, $R^2 = 0.49$. This result meant that the approximation in this study was accurate enough. Therefore, based on the SNR and the approximation accuracy, the T1 and T2 relaxation values obtained from this experiment are reasonable.

In this study, we expressed the reporter genes for MRI, transferrin receptor, DMT1, Ferritin, Ferritin-M6A, and Mms6 in cultured cells and measured the T1 and T2 relaxation values of the cells after they were exposed to metal ions to verify which reporter gene was the best for our imaging purposes. The T1 or T2 shortening effects meant that the cells uptook $MnCl_2$ or Holo-transferrin and the shorter T1 or T2 relaxation values meant that the cells uptook more metal ions.

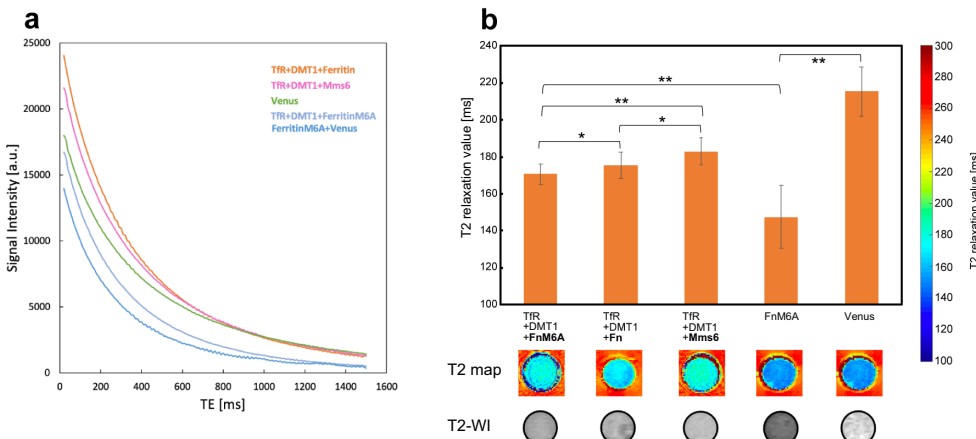

**Fig 5. T2 relaxation values of the cells expressing each gene combination when exposed to Holo-transferrin.** After the gene expression, Holo-transferrin was taken into the cells and MRI was performed with varying TE (TE = 10–1500 ms; 150 steps) (a). The T2 relaxation values are shown in the graph (*: $p < 0.05$, **: $p < 0.01$, error bar: standard deviation) and the T2 maps and T2 weighted images are shown below (b). The T2 relaxation value of the identified optimal gene (Ferritin-M6A alone) was 69% of that of the control (Venus).

Next, we will discuss the results of the T1 relaxation value measurements of the cells exposed to MnCl$_2$. First, we compared the reporter genes with roles in metal ion storage, Ferritin, Ferritin-M6A, and Mms6. Transferrin receptor and DMT1 were expressed as common terms. Among the three, the T1 relaxation value of the cells expressing transferrin receptor, DMT1, and Ferritin-M6A was the lowest (Fig 4B). Subsequently, we compared the expression of Ferritin-M6A alone and in combination with transferrin receptor and DMT1. Among these combinations, the T1 relaxation value of transferrin receptor + DMT1 + Ferritin-M6A was the lowest (Fig 4B). Based on this result, Ferritin-M6A is useful when MnCl$_2$ is used as a contrast medium. In addition, the result that the T1 relaxation value of the combination of transferrin receptor and DMT1 was lower than that of Ferritin-M6A alone was consistent with the results of previous studies [9]. Therefore, this suggested that transferrin receptor and DMT1 were more important than Ferritin-M6A when MnCl$_2$ was used. The reason for this is that Ferritin-M6A is a protein mainly involved in Fe ion retention. Therefore, Ferritin-M6A can be a useful reporter gene for MnCl$_2$ when it is simultaneously expressed with transferrin receptor and DMT1. Finally, these results suggest that the combined expression of transferrin receptor, DMT1, and Ferritin-M6A was optimal for MnCl$_2$ uptake to shorten T1.

Next, we will discuss the results of the T2 relaxation value measurement of the cells exposed to Holo-transferrin. The result that the T2 relaxation value of Ferritin-M6A was lower than that of Ferritin (Fig 5B) was consistent with the results of a previous study [13]. However, the previous study reported that the T2 relaxation value of Ferritin-M6A was 75% of that of Ferritin. In our study, the difference was smaller than this value. In addition, our study revealed that the T2 relaxation value was lower when Ferritin-M6A was expressed alone than when Ferritin-M6A, transferrin receptor, and DMT1 were expressed simultaneously (Fig 5B). These results suggest that transferrin receptor and DMT1 obstructed the T2 relaxation ability of Ferritin-M6A. Therefore, the T2 relaxation value was shortest when Ferritin-M6A was expressed alone, suggesting that Ferritin-M6A expression alone is optimal for shortening the T2 value by incorporating Holo-transferrin.

This study will contribute to improved contrast in cellular imaging using MRI. This technology, which enables imaging of targeted cells *in vivo* in a minimally invasive way, is expected to be useful in the field of regenerative medicine and other applications. When introducing differentiated cells or tissues from iPS cells, it would be very useful to know where they have grown and how they are progressing *in vivo*. In the future, we plan to shift this study from cell experiments to mouse experiments.

This study had several limitations. The cytotoxicity due to metal ion uptake was not considered in this study. This is important information to phase into animal studies. In the future, we plan to investigate the optimization of the number of metal ions to achieve a level of cytotoxicity that will not cause any problems and the maximum MRI contrast. In addition, HEK293T cells were derived from human fetal kidney cells and the gene transfer method was adapted to these cells. Therefore, the same T1 or T2 shortening effect might not be obtained when gene transfer is performed on living organisms such as mice. There are two reasons for this. First, the viral vectors, such as associated virus vector (AAV) and lentiviral vector are used for gene transfer to mice, and not lipofection. Therefore, the method of gene transfer is different. Second, it is more difficult to transduce genes to living animals by using viral vectors such as AAV, because of their packaging capacity. Although lentiviral vectors have a larger packaging capacity, they can only introduce genes into cells during mitosis. Therefore, the conditions examined in this study were based on cultured cells, and there is a possibility that different results will be obtained in living animals. However, we can say that the results of experiments in cultured cells provide the basis for evaluation *in vivo*.

## Conclusion

In this study, we investigated the best combination of reporter genes for MRI in cultured cells and obtained the following results. The combination of transferrin receptor, DMT1, and Ferritin-M6A is the best reporter gene combination for T1-weighted images using $MnCl_2$ as a contrast medium. In addition, Ferritin-M6A proved to be the best reporter gene for T2-weighted images using Holo-transferrin as a contrast medium. However, these results were only obtained in cultured cells, and the optimal genes for *in vivo* experiments in mice and other animals need to be re-examined.

## Supporting information

**S1 Raw images.**
(PDF)

## Acknowledgments

The authors would like to thank Taeko Ito for the technical assistance with the experiments and Kei Hagiya for useful discussions.

## Author Contributions

**Data curation:** Naoya Hayashi, Junichi Hata, Ayano Oku.

**Funding acquisition:** Junichi Hata, Hideyuki Okano.

**Investigation:** Naoya Hayashi, Junichi Hata, Tetsu Yoshida.

**Methodology:** Naoya Hayashi, Junichi Hata, Tetsu Yoshida, Yawara Haga, Hinako Oshiro, Noriyuki Kishi.

**Project administration:** Junichi Hata.

**Supervision:** Junichi Hata, Tetsu Yoshida, Daisuke Yoshimaru, Noriyuki Kishi, Takako Shirakawa, Hideyuki Okano.

**Validation:** Junichi Hata, Daisuke Yoshimaru.

**Visualization:** Daisuke Yoshimaru.

**Writing – original draft:** Naoya Hayashi.

**Writing – review & editing:** Junichi Hata, Daisuke Yoshimaru, Noriyuki Kishi, Hideyuki Okano.

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
