## [Decision Letter · Decision Letter 0]

16 Oct 2023

PONE-D-23-27098Identification of the reporter gene combination that shows high contrast for cellular level MRIPLOS ONE

Dear Dr. Hata,

Thank you for submitting your manuscript to PLOS ONE. After careful consideration, we feel that it has merit but does not fully meet PLOS ONE’s publication criteria as it currently stands. Therefore, we invite you to submit a revised version of the manuscript that addresses the points raised during the review process.

We look forward to receiving your revised manuscript.

Kind regards,

Kazunori Nagasaka

Academic Editor

PLOS ONE

Journal Requirements:

**Additional Editor Comments:**

Dear Authors,

Thank you very much for submission to Plos One. Overall, the manuscript is very interesting and all reviewers have commented some usuful idea to improve your manuscript.

Please revise the manuscript accordingly.

Sincerely,

Plos one editorial office

Reviewers' comments:

Reviewer's Responses to Questions

**Comments to the Author**

1. Is the manuscript technically sound, and do the data support the conclusions?

Reviewer #1: Partly

Reviewer #2: Yes

Reviewer #3: Yes

2. Has the statistical analysis been performed appropriately and rigorously? 

Reviewer #1: I Don't Know

Reviewer #2: No

Reviewer #3: Yes

3. Have the authors made all data underlying the findings in their manuscript fully available?

Reviewer #1: Yes

Reviewer #2: No

Reviewer #3: Yes

4. Is the manuscript presented in an intelligible fashion and written in standard English?

Reviewer #1: No

Reviewer #2: No

Reviewer #3: Yes

5. Review Comments to the Author

Reviewer #1: The authors tested the combination of the five genes ({TfR+DMT1} and {Fn, Fn-M6A, Mms6}) already known as MRI reporter genes to see which was the most favorable combination.

1. For me this in not a regular article, thus it should be changed to short communication.

The authors claim to have found the optimal combination, but some data and experiments are lacking.

2. The authors have adjusted the amount of DNA at transfection, but the T1 or T2 relaxation values should take into account the actual protein expression levels.

3. It may be necessary to indicate whether the combined gene is expressed as a protein in the same cell. It is also necessary to see that the expression levels do not change depending on the combination. For example, it is necessary to see if the expression level of TfR+DMT1 is altered by co-transfection of the partner {Fn, Fn-M6A, Mms6, Venus}.

4. I'm not a specialist so I don't know the details, but is it not necessary to measure both T1 and T2 relaxation values in each sample to which the metal ion MnCl2 or Holo-transferrin have been added?

Even when they found the best combination of them, I thought the impact was low in terms of how they contributed to the improvement of reporter MRI.

5. In fact, when I looked at the graphs in figure4,5, I saw only a slight improvement in that value. I am not sure how favorable this change in value is. I think the authors should provide an explanation as well as a specific example such as something that was undetectable became detectable for the first time with the combination the authors obtained.

Other points:

6. The title says "high contrast," but it is only looking at the relaxation value of T1 and T2. To say this, MRI images of a mixture of cells with and without reporter genes must be produced, and visually and quantitatively convincing data must be obtained to show that the combination found improved the detection of cells expressing the reporter gene with high contrast.

7.The title says cellular level, but isn't this a term for detection at the level of a single cell? I assume that this experiment only analyzes cell populations collected from 6-well dishes.

8.The authors state in the abstract and introduction that GFP is inappropriate for use in the observation of living cells (in vivo imaging), but could the argument be applied to other fluorescent proteins as well? Shouldn't the authors describe fluorescent proteins rather than GFP?

9. Each of Fig4 and 5 may show the analyzed samples of Table1 measured in Fig3d and e, so they can be merged into one graph. As it is now, some bars are shown in duplicate.

10. The legend of Fig.3 precedes that of Fig.2.

11.Since PCR was not performed, it is better to use expression such as 0.2mL tube instead of PCR tube.

12.Some parts of the last paragraph are missing, along with garbled text.

13. line 156: This statement needs to be modified because FLAG tags can affect function depending on the protein and fusion position.

Reviewer #2: The study suggested that combining two or more MRI reporters would provide better contrast and resolution during application. The authors are recommended to revise the following items.

1. The structure of the current manuscript is not standardized. For example, figure descriptions were mixed with methodologies in the method part. Figure titles are used as subtitles in many paragraphs.

2. A description on statistical analysis is missing in the methods and results section. The authors are also recommended to report effect size statistics in the results section, as the difference between groups appears to be small and the error bar is large. Meanwhile, the authors did not report what the error bars represent. Are they SD or SEM?

3. I have concerns about the high cytotoxicity (neurotoxicity) of reporter cocktail due to the accumulation of high levels of iron. This is important for long-term study. For example, to observe the changes in the structure of the brain regions, it may take more than a month. Please discuss this point.

4. The difficulty in in vivo transduction lies in the packaging capacity of AAV vector, but not in the transduction efficiency of AAV (Lines 262-263 are not appropriate). In this regard, lentiviral vectors with much bigger packaging capacity are an option. Please discuss this point.

5. Please consider rearranging the figure panels.

6. Figure 2: Can the authors compare the expression levels of five reporters between different combinations?

Reviewer #3: In the manuscript “Identification of the reporter gene combination that shows high contrast for cellular level MRI”, by Naoya Hayashi, Junichi Hata, Tetsu Yoshida, Daisuke Yoshimaru, Yawara Haga, Hinako Ohshiro, Noriyuki Kishi, Takako Shirakawa, Hideyuki Okano , authors provide new insights genes, which allow the MRI-mediated observation of labeled cells. This is undoubtedly an interesting study that expands knowledge about the new approaches for visualizing cells potentially in vivo and will be of interest to a wide range of researchers.

Although the study is of undoubted interest and expands our knowledge of new methodological approaches in the study of cells to increase the level of validity of the material and publication in the journal, it needs some improvements.

The entire study was performed on HEK293TB cell culture. But this is not reflected neither in the title nor in the abstract. The abstract says: " Therefore, in this study, we examined the reporter genes, which allow the MRI-mediated observation of labeled cells in living animals.". Potentially these genes can be used in in vivo studies, but this study was performed on cell culture and this needs to be reflected in the abstract.

Some aspects of the conducted study are quite well reflected in the methodology, but some points are missing. Fig. 1 shows microphotographs of cells, but there is no scale bar. It is necessary to add a scale bar. It is also necessary to add the description of used microscopic equipment to the methodology. Microscope model, lenses, filters, wavelengths at which fluorescence was excited, etc.

Figure 1 on the left shows a micrograph of cells in bright fild mode on the right overlaying bright fild and the GFP channel. It is necessary to reflect this in the caption to the figure. It is not quite clear why the GFP channel is presented and how it relates to the study performed. I did not find any information either in the methods or in the results. It is necessary to explain it.

It is also necessary to describe in more detail the equipment used for MRI experiments. The phrase "We used a 9.4 T ultrahigh field MRI (Bruker, Biospin, Ettlingen, Germany) in the 133 RIKEN Center for Brain Science." is not sufficient. It is necessary to specify the model of the equipment.

In the discussion, the authors mainly describe the results obtained and their statistical significance. There is a lack of comparison with the data obtained earlier. What are the advantages of the methodology proposed by the authors compared to what has been done before. Moreover, in the introduction the authors refer to the works performed earlier. It means it is possible to compare the data with those obtained earlier.

How do the authors plan to transfer the obtained approaches to in vivo work? This also needs to be discussed. And how the existing experimental approach can be applied as it is presented in this paper. On a single cell culture. What could potentially be studied with it. If there are problems with transferring this technique to the whole animal as the authors claim. The discussion needs to be expanded, especially since there is much to discuss.

With all the suggested additions, the paper can certainly be accepted for publication.

6. PLOS authors have the option to publish the peer review history of their article (what does this mean?). If published, this will include your full peer review and any attached files.

Reviewer #1: No

Reviewer #2: No

Reviewer #3: No

---

## [Author Response · Author response to Decision Letter 0]

19 Dec 2023

Dear Editor and Reviewers: 

We wish to re-submit the manuscript titled “Identification of the reporter gene combination that shows high contrast for cellular level MRI.” The manuscript ID is PONE-D-23-27098.

We thank you and the reviewers for your thoughtful suggestions and insights. The manuscript has benefited from these insightful suggestions. I look forward to working with you and the reviewers to move this manuscript closer to publication in PLOS ONE.

I would like to inform you that during the revision process, a new author, Ayano Oku, has been added to the manuscript. The new author helped us with additional experiments and considerations.

The manuscript has been rechecked and the necessary changes have been made in accordance with the reviewers’ suggestions. The modified sections are indicated in red in “Revised Manuscript with Track Changes” file. In addition, our manuscripts have been proofread in English to improve quality, and all corrections are highlighted also in “Revised Manuscript with Track Changes” file. The responses to all comments have been prepared and given below.

Dear Editor

>>> 1. Please ensure that your manuscript meets PLOS ONE's style requirements, including those for file naming.

Thank you for your comment. We have submitted our manuscript in accordance with PLOS ONE's style requirements.

>>> 2. We suggest you thoroughly copyedit your manuscript for language usage, spelling, and grammar. If you do not know anyone who can help you do this, you may wish to consider employing a professional scientific editing service. 

Thank you for your important point. We had our manuscript proofread in English. As noted above, we have highlighted the parts of the corrections in the “supporting information” file. The name of scientific editing service is “editage”.

>>> 3. In your Data Availability statement, you have not specified where the minimal data set underlying the results described in your manuscript can be found. PLOS defines a study's minimal data set as the underlying data used to reach the conclusions drawn in the manuscript and any additional data required to replicate the reported study findings in their entirety. All PLOS journals require that the minimal data set be made fully available. For more information about our data policy, please see http://journals.plos.org/plosone/s/data-availability.

Thank you for telling us about the minimal data set. We uploaded our minimal data set to the following URL: https://drive.google.com/drive/folders/1KB9fq44es2qBJ488P5c_LVUo5Vfk6YAf?usp=drive_link

>>> 4. PLOS ONE now requires that authors provide the original uncropped and unadjusted images underlying all blot or gel results reported in a submission’s figures or Supporting Information files. This policy and the journal’s other requirements for blot/gel reporting and figure preparation are described in detail at https://journals.plos.org/plosone/s/figures#loc-blot-and-gel-reporting-requirements and https://journals.plos.org/plosone/s/figures#loc-preparing-figures-from-image-files. When you submit your revised manuscript, please ensure that your figures adhere fully to these guidelines and provide the original underlying images for all blot or gel data reported in your submission. See the following link for instructions on providing the original image data: https://journals.plos.org/plosone/s/figures#loc-original-images-for-blots-and-gels.

Thank you for telling us about the rule of the blot or gel results. We uploaded our uncropped Western blotting data to the following URL: https://drive.google.com/drive/folders/1KB9fq44es2qBJ488P5c_LVUo5Vfk6YAf?usp=drive_link

>>> 5. PLOS requires an ORCID iD for the corresponding author in Editorial Manager on papers submitted after December 6th, 2016. Please ensure that you have an ORCID iD and that it is validated in Editorial Manager. To do this, go to ‘Update my Information’ (in the upper left-hand corner of the main menu), and click on the Fetch/Validate link next to the ORCID field. This will take you to the ORCID site and allow you to create a new iD or authenticate a pre-existing iD in Editorial Manager. Please see the following video for instructions on linking an ORCID iD to your Editorial Manager account: https://www.youtube.com/watch?v=_xcclfuvtxQ

Thank you for telling us about ORCID ID. We linked the first author’s and corresponding author’s ORCID ID.

>>> 6. Please review your reference list to ensure that it is complete and correct. If you have cited papers that have been retracted, please include the rationale for doing so in the manuscript text, or remove these references and replace them with relevant current references. Any changes to the reference list should be mentioned in the rebuttal letter that accompanies your revised manuscript. If you need to cite a retracted article, indicate the article’s retracted status in the References list and also include a citation and full reference for the retraction notice.

Thank you for your comments. There are no retractions or changes to the reference list in this revision.

Dear Reviewer 1

>>> 1. For me this in not a regular article, thus it should be changed to short communication.

Thank you for your comment. We have added more data and considering other reviewers' comments as well, we would like to submit it as a regular article.

>>> 2. The authors have adjusted the amount of DNA at transfection, but the T1 or T2 relaxation values should take into account the actual protein expression levels. 

3. It may be necessary to indicate whether the combined gene is expressed as a protein in the same cell. It is also necessary to see that the expression levels do not change depending on the combination. For example, it is necessary to see if the expression level of TfR+DMT1 is altered by co-transfection of the partner {Fn, Fn-M6A, Mms6, Venus}.

Thank you for your comment. The expression levels when co-expressed were essential to the discussion. I have added the Western blot results when co-expressed in Fig 2. In each combination, the Western blot concentration of each protein did not decrease. From the added figure, we can say that there was no change in the expression levels due to the combination.

>>> 4. I'm not a specialist so I don't know the details, but is it not necessary to measure both T1 and T2 relaxation values in each sample to which the metal ion MnCl2 or Holo-transferrin have been added?

Thank you for your important point. We have reported the shortening of T1 relaxation values for Mn ions* and the shortening of T2 relaxation values for Fe ions in Holo-transferrin**, and we have focused on these two aspects in this study.

*Nordhøy W, et al. Intracellular manganese ions provide strong T1 relaxation in rat myocardium. Magn Reson Med. 2004;52(3):506-514.

**Aslan TN. Relaxivity properties of magnetoferritin: The iron loading effect. J Biosci Bioeng. 2022;133(5):474-480.

>>> 5. In fact, when I looked at the graphs in figure 4,5, I saw only a slight improvement in that value. I am not sure how favorable this change in value is. I think the authors should provide an explanation as well as a specific example such as something that was undetectable became detectable for the first time with the combination the authors obtained.

Thank you for highlighting this. We agree with your statement that the difference between T1 and T2 relaxation values is very small. In MRI, this small difference between T1 and T2 relaxation values can be depicted as a contrast on the image. For example, as shown in the following paper*, the difference between T1 and T2 relaxation values in each region of the brain is small; however, contrast can be obtained by using different imaging conditions. In other words, even slight improvement is very important to increase the detection of cells in MRI.

*Zheng Z, Liu Y, Yin H, et al. Evaluating T1, T2 Relaxation, and Proton Density in Normal Brain Using Synthetic MRI with Fast Imaging Protocol [published online ahead of print, 2023 Sep 8]. Magn Reson Med Sci. 2023;10.2463/mrms.tn.2022-0161.

>>> 6. The title says "high contrast," but it is only looking at the relaxation value of T1 and T2. To say this, MRI images of a mixture of cells with and without reporter genes must be produced, and visually and quantitatively convincing data must be obtained to show that the combination found improved the detection of cells expressing the reporter gene with high contrast.

As you have pointed out, it was difficult to understand how much the detection ability differs when actually imaging. We have added the T1-weighted image (T1-WI) and T2-weighted image (T2-WI) to Figures 4 and 5 to validate the detection ability visually. We also believe that quantitative validation can be explained by the T1 and T2 relaxation values.

>>> 7. The title says cellular level, but isn't this a term for detection at the level of a single cell? I assume that this experiment only analyzes cell populations collected from 6-well dishes.

Thank you for your comment. In this study, we only imaged cells collected from 6-well dishes. In MRI, the signal values of substances in a voxel are averaged to generate an image of a single voxel. This means that if there is even one high-signal substance in a voxel, that voxel will have a high signal. This is called the partial volume effect, which allows us to detect even a single cell containing metal ions. In recent years, it has become possible to improve the detection performance by correcting the image to account for this partial volume effect*.

*Espe EKS, Bendiksen BA, Zhang L, Sjaastad I. Analysis of right ventricular mass from magnetic resonance imaging data: a simple post-processing algorithm for correction of partial-volume effects. Am J Physiol Heart Circ Physiol. 2021;320(2):H912-H922.

>>> 8. The authors state in the abstract and introduction that GFP is inappropriate for use in the observation of living cells (in vivo imaging), but could the argument be applied to other fluorescent proteins as well? Shouldn't the authors describe fluorescent proteins rather than GFP?

Thank you for your comment. The description in the first half of the Introduction was applicable not only to GFP but also to fluorescent proteins in general. The revised parts are as follows:

For example, fluorescent protein such as green fluorescence protein is commonly used for cell labeling. However, fluorescent protein is difficult to observe in living animals. (Line 29-30, Page 2)

>>> 9. Each of Fig4 and 5 may show the analyzed samples of Table1 measured in Fig3d and e, so they can be merged into one graph. As it is now, some bars are shown in duplicate.

Thank you for your comment. We agree with you that the elements of the graphs were duplicated, making the Figure difficult to read. We have merged the bar graphs in Figures 4 and 5 into a single graph so that the bar graphs do not overlap.

>>> 10. The legend of Fig.3 precedes that of Fig.2.

Thank you for your comment. As highlighted, we have swapped the order of Figures 2 and 3 accordingly.

>>> 11.Since PCR was not performed, it is better to use expression such as 0.2mL tube instead of PCR tube.

Thank you for your comment. It was unnatural to describe the tubes as PCR tube when the PCR method was not used. We have replaced “PCR tube” with “0.2 mL tubes with a pointed tip.” (Line 134-135, 151)

>>> 12.Some parts of the last paragraph are missing, along with garbled text.

Thank you for highlighting this and we apologize for the missing parts. We have corrected the missing parts as follows:

However, we can say that the results of experiments in cultured cells provide the basis for evaluation in vivo. (Line 282-283, Page 11)

>>> 13. line 156: This statement needs to be modified because FLAG tags can affect function depending on the protein and fusion position.

We agree with your comment about the description of the FLAG tag. We have revised the description to state that there is no effect when fused to the appropriate position of the appropriate protein. The corrected part is as follows:

FLAG tag is known to have no effect on protein function if fused to the appropriate position on the appropriate protein [19]. (Line 162, Page 7)

Dear Reviewer 2

>>> 1. The structure of the current manuscript is not standardized. For example, figure descriptions were mixed with methodologies in the method part. Figure titles are used as subtitles in many paragraphs.

Thank you for your comment. As highlighted, there was a duplication of content, which made the manuscript difficult to read.

We have removed the overlap between the captions of Figures 2 and 3 and the text of the Figure so that only one of them is used as much as possible (Captions of Figures 2 and 3, Page 6-7).

>>> 2. A description on statistical analysis is missing in the methods and results section. The authors are also recommended to report effect size statistics in the results section, as the difference between groups appears to be small and the error bar is large. Meanwhile, the authors did not report what the error bars represent. Are they SD or SEM?

Thank you for your comments on the statistical analysis. We have added the following description of statistical analysis to the Material and Method section:

The statistical significance test was performed using analysis of variance and Bonferroni methods. The significance level was set at p = 0.05. (Line 145-147, Page 6)

We agree with you that the effect size should be described in addition to the p-value. In this case, since n is small, calculating the effect size would inevitably result in a large value, and there is a possibility of overestimating the significance of the difference; therefore, we decided to include only the p-value.

As for the error bars in Figures 4 and 5, we added the following note that they indicate standard deviation:

The T1 relaxation values are shown in the graph (*: p < 0.05, **: p < 0.01, error bar: standard deviation) (Line 190-191, Page 8)

The T2 relaxation values are shown in the graph (*: p < 0.05, **: p < 0.01, error bar: standard deviation) (Line 200-201, Page 8)

>>> 3. I have concerns about the high cytotoxicity (neurotoxicity) of reporter cocktail due to the accumulation of high levels of iron. This is important for long-term study. For example, to observe the changes in the structure of the brain regions, it may take more than a month. Please discuss this point.

We believe that cytotoxicity is an issue that should be considered, as you have pointed out. However, since it is currently difficult to make an additional study on this issue, we have added the following to the Limitation section:

Thank you for your comment. Cytotoxicity due to metal ion uptake was not considered in this study. This is important information to phase into animal studies. In the future, we plan to investigate the optimization of the number of metal ions to achieve a level of cytotoxicity that will not cause any problems and the maximum MRI contrast. (Line 270-273, Page 11)

>>> 4. The difficulty in in vivo transduction lies in the packaging capacity of AAV vector, but not in the transduction efficiency of AAV (Lines 262-263 are not appropriate). In this regard, lentiviral vectors with much bigger packaging capacity are an option. Please discuss this point.

Thank you for your comment about the characteristics of the AAV. We agree with you and have revised the description of AAV accordingly. We have also added the comparison between lentiviral vectors and AAV. The corrections are as follows:

There are two reasons for this. First, the viral vectors, such as associated virus vector (AAV) and lentiviral vector, are used for gene transfer to mice, and not lipofection. Therefore, the method of gene transfer is different. Second, it is more difficult to transduce genes to living animals by using viral vectors, such as AAV, because of their packaging capacity. Although lentiviral vectors have a larger packaging capacity, they can only introduce genes into cells during mitosis. (Line 276-280, Page 11)

>>> 5. Please consider rearranging the figure panels.

Thank you for your comments about the Figure panels. We have rearranged and modified the Figures based on the feedback from other reviewers. Specifically, the revised sections are as follows:

Figure 1: We have added scale bars to the micrographs of the cells.

Figure 2: We have added another Western blot results for co-expression of the reporter gene.

Figure 3: We have moved the relaxation curves to Figures 4 and 5.

Figure 4: We have moved the T1 relaxation curve from Figure 3 to Figure 4. We have also unified the three bar charts into one.

Figure 5: We have moved the T2 relaxation curve from Figure 3 to Figure 5. We have also unified the three bar charts into one.

>>> 6. Figure 2: Can the authors compare the expression levels of five reporters between different combinations?

Thank you for your comment. The expression level of the protein when co-expressed was essential for consideration. I have added another Western blot result of the combined expression of the reporter genes in Figure 2. This allowed us to compare the expression levels of the reporter genes in different combinations.

Dear Reviewer 3

>>> The entire study was performed on HEK293TB cell culture. But this is not reflected neither in the title nor in the abstract. The abstract says: " Therefore, in this study, we examined the reporter genes, which allow the MRI-mediated observation of labeled cells in living animals.". Potentially these genes can be used in in vivo studies, but this study was performed on cell culture and this needs to be reflected in the abstract.

Thank you for your comment. The information highlighted by you was missing from the abstract. In the abstract, we have stated that cell culture was targeted as a preliminary study for animal experiments as follows:

As a preliminary stage of animal study, we transduced some groups of plasmids that coded the protein that could take and store metal ions to the cell culture, added metal ions solutions, and measured their T1 or T2 relaxation values. (Line 33-34, Page 2)

>>> Some aspects of the conducted study are quite well reflected in the methodology, but some points are missing. Fig. 1 shows microphotographs of cells, but there is no scale bar. It is necessary to add a scale bar. It is also necessary to add the description of used microscopic equipment to the methodology. Microscope model, lenses, filters, wavelengths at which fluorescence was excited, etc.

Thank you for pointing out the missing information regarding the equipment used. Since there was no description, we added the scale bar of the micrograph to Figure 1 in the Material and Method section and added the following description about the model of microscope, lens, filter, and wavelength of fluorescence in the Material and Method section.

The microscope information is as follows: BZ-X710 (KEYENCE, Osaka, Japan), magnification, x10, phase contrast objective lens, BZ-PF10P (KEYENCE); fluorescent filter for GFP; OP-87762 (excitation wavelength = 360/40 nm, absorption wavelength = 460/50 nm, dichroic mirror wavelength = 400 nm, KEYENCE). (Line 78-81, Page 3-4)

>>> Figure 1 on the left shows a micrograph of cells in bright field mode on the right overlaying bright field and the GFP channel. It is necessary to reflect this in the caption to the figure. It is not quite clear why the GFP channel is presented and how it relates to the study performed. I did not find any information either in the methods or in the results. It is necessary to explain it.

Thank you for your comments regarding the lack of explanation regarding Figure 1. We have corrected the caption of Figure 1 accordingly.

Figure 1 shows that GFP is being expressed in the cells, indicating that the cells are not strange and are not dead. We have also included it to visually show that the gene transfer was successful. This explanation has been added to the caption of Figure 1 as follows:

Left: microscopic image of HEK293T cell in the bright field mode; Right: microscopic image of HEK293T cells expressing GFP, overlaying bright field and the GFP channel (a). Microscopic observation of GFP expression in the cells confirmed that they were not strange cells and were not dead. (Caption of Figure 1, Line 105-108, Page 5)

>>> It is also necessary to describe in more detail the equipment used for MRI experiments. The phrase "We used a 9.4 T ultrahigh field MRI (Bruker, Biospin, Ettlingen, Germany) in the 133 RIKEN Center for Brain Science." is not sufficient. It is necessary to specify the model of the equipment.

Thank you for your comment. We have added information about MRI to the Material and Method section. Specifically, the MRI model name and Max Gradient strength have been added as shown below:

We used a 9.4 T ultrahigh field MRI (Bruker, Biospin94/30, Max Gradient Strength 660 mT/m, Ettlingen, Germany) in the RIKEN Center for Brain Science. (Line 138-139, Page 6)

>>> In the discussion, the authors mainly describe the results obtained and their statistical significance. There is a lack of comparison with the data obtained earlier. What are the advantages of the methodology proposed by the authors compared to what has been done before. Moreover, in the introduction the authors refer to the works performed earlier. It means it is possible to compare the data with those obtained earlier.

We agree with you that it is important to make a comparison of the values with previous studies. However, in this study, it was difficult to simply compare the values because the cells and MRI conditions used were different from those used in previous studies. Therefore, we introduced genes as closely as possible to the previous studies, expressed proteins, and compared the results to determine the optimal reporter gene.

>>> How do the authors plan to transfer the obtained approaches to in vivo work? This also needs to be discussed. And how the existing experimental approach can be applied as it is presented in this paper. On a single cell culture. What could potentially be studied with it. If there are problems with transferring this technique to the whole animal as the authors claim. The discussion needs to be expanded, especially since there is much to discuss.

We agree with you that we should describe the future prospects of this project. We have added to the discussion that this technology has the potential to play an active role in the field of regenerative medicine and that in order to realize this potential, this study will be shifted to animal experiments in the future. Specifically, the revised sections are as follows:

This study will contribute to improved contrast in cellular imaging by MRI. This technology, which enables imaging of targeted cells in vivo in a minimally invasive way, is expected to be useful in the field of regenerative medicine and other applications. When introducing differentiated cells or tissues from iPS cells, it would be very useful to know where they have grown and how they are progressing in vivo. In the future, we plan to shift this study from cell experiments to mouse experiments. (Line 264-269, Page 11)

Thank you again for your valuable comments and suggestions and for giving us the opportunity to strengthen our manuscript. We have worked hard to incorporate your input and hope that these corrections will make you more receptive to our submission.]

Sincerely,

Junichi Hata

Tokyo Metropolitan University

---

## [Editor Report · Decision Letter 1]

26 Dec 2023

PONE-D-23-27098R1Identification of the reporter gene combination that shows high contrast for cellular level MRIPLOS ONE

Dear Dr. Hata,

Thank you for submitting your manuscript to PLOS ONE. After careful consideration, we feel that it has merit but does not fully meet PLOS ONE’s publication criteria as it currently stands. Therefore, we invite you to submit a revised version of the manuscript that addresses the points raised during the review process.

We look forward to receiving your revised manuscript.

Kind regards,

Kazunori Nagasaka

Academic Editor

PLOS ONE

Journal Requirements:

Additional Editor Comments:

Dear Authors,

Thank you very much for your submission to Plos One.

Our expert reviewers have commented to your manuscript and overall I think the content is very informative and useful.

Our decision is minor revision and we look forward your revised manuscript soon.

Sincerely,

Plos One

---

## [Author Response · Author response to Decision Letter 1]

28 Dec 2023

Dear Editor and Reviewers: 

We wish to re-submit the manuscript titled “Identification of the reporter gene combination that shows high contrast for cellular level MRI.” The manuscript ID is PONE-D-23-27098R1.

We thank you and the reviewers for your thoughtful comments. The manuscript has benefited from your suggestions, and I would like to address your comments in the following manner. Once again, I am grateful for your valuable insights.

The reference section in the manuscript has been rechecked and the necessary changes have been made in accordance with your suggestions. We hope we have met your intentions. The modified sections are highlighted in “Revised Manuscript with Track Changes” file.

Dear Editor

>>> Please review your reference list to ensure that it is complete and correct. If you have cited papers that have been retracted, please include the rationale for doing so in the manuscript text, or remove these references and replace them with relevant current references. Any changes to the reference list should be mentioned in the rebuttal letter that accompanies your revised manuscript. If you need to cite a retracted article, indicate the article’s retracted status in the References list and also include a citation and full reference for the retraction notice.

Thank you for your comment. We rechecked our reference list and corrected them. We have not removed or added references, only modified some statements. The corrected reference list is as follows:

1. Blasberg RG, Tjuvajev JG. Molecular-genetic imaging: current and future perspectives. J Clin Invest. 2003;111(11): 1620–1629. doi:10.1172/JCI18855

2. Shimomura O. Discovery of green fluorescent protein (GFP) (Nobel Lecture). Angew Chem Int Ed Engl. 2009;48(31): 5590–5602. doi:10.1002/anie.200902240

3. Iyer S, Arindkar S, Mishra A, Manglani K, Kumar JM, Majumdar SS, et al. Development and Evaluation of Transgenic Nude Mice Expressing Ubiquitous Green Fluorescent Protein. Mol Imaging Biol. 2015;17(4): 471–478. doi:10.1007/s11307-014-0821-5

4. Hall MP, Unch J, Binkowski BF, Valley MP, Butler BL, Wood MG, et al. Engineered luciferase reporter from a deep sea shrimp utilizing a novel imidazopyrazinone substrate. ACS Chem Biol. 2012;7(11): 1848–1857. doi:10.1021/cb3002478.

5. Zarychta-Wiśniewska W, Burdzinska A, Zagozdzon R, Dybowski B, Butrym M, Gajewski Z, et al. In vivo imaging system for explants analysis-A new approach for assessment of cell transplantation effects in large animal models. PLOS ONE. 2017;12(9): e0184588. doi:10.1371/journal.pone.0184588

6. Li Y, Jiao Q, Xu H, Du X, Shi L, Jia F, et al. Biometal Dyshomeostasis and Toxic Metal Accumulations in the Development of Alzheimer’s Disease. Front Mol Neurosci. 2017;10: 339. doi:10.3389/fnmol.2017.00339

7. Aisen P. Transferrin receptor 1. Int J Biochem Cell Biol. 2004;36(11): 2137–2143. doi:10.1016/j.biocel.2004.02.007

8. Andrews NC. The iron transporter DMT1. Int J Biochem Cell Biol. 1999;31(10): 991–994. doi:10.1016/S1357-2725(99)00065-5

9. Lewis CM, Graves SA, Hernandez R, Valdovinos HF, Barnhart TE, Cai W, et al. 52Mn production for PET/MRI tracking of human stem cells expressing divalent metal transporter 1 (DMT1). Theranostics. 2015;5(3): 227–239. doi:10.7150/thno.10185

10. Ono K, Fuma K, Tabata K, Sawada M. Ferritin reporter used for gene expression imaging by magnetic resonance. Biochem Biophys Res Commun. 2009;388(3): 589–594. doi:10.1016/j.bbrc.2009.08.055

11. Yang C, Tian R, Liu T, Liu G. MRI Reporter Genes for Noninvasive Molecular Imaging. Molecules. 2016;21(5):580. doi:10.3390/molecules21050580

12. Galloway JM, Arakaki A, Masuda F, Tanaka T, Matsunaga T, Staniland SS. Magnetic bacterial protein Mms6 controls morphology, crystallinity and magnetism of cobalt-doped magnetite nanoparticles in vitro. J. Mater. Chem., 2011;21: 15244-15254. doi:10.1039/C1JM12003D

13. Graham FL, Smiley J, Russell WC, Nairn R. Characteristics of a human cell line transformed by DNA from human adenovirus type 5. J Gen Virol. 1977;36(1): 59–74. doi:10.1099/0022-1317-36-1-59

14. Yuan J, Xu WW, Jiang S, Yu H, Fai Poon HF. The Scattered Twelve Tribes of HEK293. Biomed Pharmacol J. 2018;11(2):621–623. Available from: http://biomedpharmajournal.org/?p=20696

15. Radoul M, Lewin L, Cohen B, Oren R, Popov S, Davidov G, et al. Genetic manipulation of iron biomineralization enhances MR relaxivity in a ferritin-M6A chimeric complex. Sci Rep. 2016;6: 26550. doi:10.1038/srep26550

16. Kurien BT, Scofield RH. Western blotting: an introduction. Methods Mol Biol. 2015;1312: 17–30. doi:10.1007/978-1-4939-2694-7_5

17. Towbin H, Staehelin T, Gordon J. Electrophoretic transfer of proteins from polyacrylamide gels to nitrocellulose sheets: procedure and some applications. Proc Natl Acad Sci U S A. 1979;76(9): 4350–4354. doi:10.1073/pnas.76.9.4350

18. Dou Y, Lin Y, Wang TY, Wang XY, Jia YL, Zhao CP. The CAG promoter maintains high-level transgene expression in HEK293 cells. FEBS Open Bio. 2021;11(1): 95–104. doi:10.1002/2211-5463.13029

19. Einhauer A, Jungbauer A. The FLAG peptide, a versatile fusion tag for the purification of recombinant proteins. J Biochem Biophys Methods. 2001;49(1-3): 455–465. doi:10.1016/s0165-022x(01)00213-5

20. Dohi T, Murashige K. A Micro Saddle Coil with Switchable Sensitivity for Local High-Resolution Imaging of Luminal Tissue. Micromachines. 2016;7(4): 67. doi:10.3390/mi7040067

Thank you again for your valuable comments and for giving us the opportunity to improve our manuscript. We have worked hard to incorporate your input and hope that these corrections will make you more receptive to our submission.

Sincerely,

Corresponding Auther:

Junichi Hata

Phone: +81-03-3819-1211 (extension; 410), Email: j-hata@tmu.ac.jp, 

Graduate School of Human Health Sciences, Tokyo Metropolitan University, 

7-2-10 Higashi-Ogu, Arakawa-ku, Tokyo, Japan 116-8551

First Auther:

Naoya Hayashi

Department of Radiological Sciences, Graduate School of Human Health Sciences, Tokyo Metropolitan University

7-2-10 Higashi-Ogu, Arakawa-ku, Tokyo, Japan 116-8551

Tel: 03-3819-1211

E-mail: n.hayashi731@gmail.com

Dear Editor and Reviewers: 

We wish to re-submit the manuscript titled “Identification of the reporter gene combination that shows high contrast for cellular level MRI.” The manuscript ID is PONE-D-23-27098R1.

We thank you and the reviewers for your thoughtful comments. The manuscript has benefited from your suggestions, and I would like to address your comments in the following manner. Once again, I am grateful for your valuable insights.

The reference section in the manuscript has been rechecked and the necessary changes have been made in accordance with your suggestions. We hope we have met your intentions. The modified sections are highlighted in “Revised Manuscript with Track Changes” file.

Dear Editor

>>> Please review your reference list to ensure that it is complete and correct. If you have cited papers that have been retracted, please include the rationale for doing so in the manuscript text, or remove these references and replace them with relevant current references. Any changes to the reference list should be mentioned in the rebuttal letter that accompanies your revised manuscript. If you need to cite a retracted article, indicate the article’s retracted status in the References list and also include a citation and full reference for the retraction notice.

Thank you for your comment. We rechecked our reference list and corrected them. We have not removed or added references, only modified some statements. The corrected reference list is as follows:

1. Blasberg RG, Tjuvajev JG. Molecular-genetic imaging: current and future perspectives. J Clin Invest. 2003;111(11): 1620–1629. doi:10.1172/JCI18855

2. Shimomura O. Discovery of green fluorescent protein (GFP) (Nobel Lecture). Angew Chem Int Ed Engl. 2009;48(31): 5590–5602. doi:10.1002/anie.200902240

3. Iyer S, Arindkar S, Mishra A, Manglani K, Kumar JM, Majumdar SS, et al. Development and Evaluation of Transgenic Nude Mice Expressing Ubiquitous Green Fluorescent Protein. Mol Imaging Biol. 2015;17(4): 471–478. doi:10.1007/s11307-014-0821-5

4. Hall MP, Unch J, Binkowski BF, Valley MP, Butler BL, Wood MG, et al. Engineered luciferase reporter from a deep sea shrimp utilizing a novel imidazopyrazinone substrate. ACS Chem Biol. 2012;7(11): 1848–1857. doi:10.1021/cb3002478.

5. Zarychta-Wiśniewska W, Burdzinska A, Zagozdzon R, Dybowski B, Butrym M, Gajewski Z, et al. In vivo imaging system for explants analysis-A new approach for assessment of cell transplantation effects in large animal models. PLOS ONE. 2017;12(9): e0184588. doi:10.1371/journal.pone.0184588

6. Li Y, Jiao Q, Xu H, Du X, Shi L, Jia F, et al. Biometal Dyshomeostasis and Toxic Metal Accumulations in the Development of Alzheimer’s Disease. Front Mol Neurosci. 2017;10: 339. doi:10.3389/fnmol.2017.00339

7. Aisen P. Transferrin receptor 1. Int J Biochem Cell Biol. 2004;36(11): 2137–2143. doi:10.1016/j.biocel.2004.02.007

8. Andrews NC. The iron transporter DMT1. Int J Biochem Cell Biol. 1999;31(10): 991–994. doi:10.1016/S1357-2725(99)00065-5

9. Lewis CM, Graves SA, Hernandez R, Valdovinos HF, Barnhart TE, Cai W, et al. 52Mn production for PET/MRI tracking of human stem cells expressing divalent metal transporter 1 (DMT1). Theranostics. 2015;5(3): 227–239. doi:10.7150/thno.10185

10. Ono K, Fuma K, Tabata K, Sawada M. Ferritin reporter used for gene expression imaging by magnetic resonance. Biochem Biophys Res Commun. 2009;388(3): 589–594. doi:10.1016/j.bbrc.2009.08.055

11. Yang C, Tian R, Liu T, Liu G. MRI Reporter Genes for Noninvasive Molecular Imaging. Molecules. 2016;21(5):580. doi:10.3390/molecules21050580

12. Galloway JM, Arakaki A, Masuda F, Tanaka T, Matsunaga T, Staniland SS. Magnetic bacterial protein Mms6 controls morphology, crystallinity and magnetism of cobalt-doped magnetite nanoparticles in vitro. J. Mater. Chem., 2011;21: 15244-15254. doi:10.1039/C1JM12003D

13. Graham FL, Smiley J, Russell WC, Nairn R. Characteristics of a human cell line transformed by DNA from human adenovirus type 5. J Gen Virol. 1977;36(1): 59–74. doi:10.1099/0022-1317-36-1-59

14. Yuan J, Xu WW, Jiang S, Yu H, Fai Poon HF. The Scattered Twelve Tribes of HEK293. Biomed Pharmacol J. 2018;11(2):621–623. Available from: http://biomedpharmajournal.org/?p=20696

15. Radoul M, Lewin L, Cohen B, Oren R, Popov S, Davidov G, et al. Genetic manipulation of iron biomineralization enhances MR relaxivity in a ferritin-M6A chimeric complex. Sci Rep. 2016;6: 26550. doi:10.1038/srep26550

16. Kurien BT, Scofield RH. Western blotting: an introduction. Methods Mol Biol. 2015;1312: 17–30. doi:10.1007/978-1-4939-2694-7_5

17. Towbin H, Staehelin T, Gordon J. Electrophoretic transfer of proteins from polyacrylamide gels to nitrocellulose sheets: procedure and some applications. Proc Natl Acad Sci U S A. 1979;76(9): 4350–4354. doi:10.1073/pnas.76.9.4350

18. Dou Y, Lin Y, Wang TY, Wang XY, Jia YL, Zhao CP. The CAG promoter maintains high-level transgene expression in HEK293 cells. FEBS Open Bio. 2021;11(1): 95–104. doi:10.1002/2211-5463.13029

19. Einhauer A, Jungbauer A. The FLAG peptide, a versatile fusion tag for the purification of recombinant proteins. J Biochem Biophys Methods. 2001;49(1-3): 455–465. doi:10.1016/s0165-022x(01)00213-5

20. Dohi T, Murashige K. A Micro Saddle Coil with Switchable Sensitivity for Local High-Resolution Imaging of Luminal Tissue. Micromachines. 2016;7(4): 67. doi:10.3390/mi7040067

Thank you again for your valuable comments and for giving us the opportunity to improve our manuscript. We have worked hard to incorporate your input and hope that these corrections will make you more receptive to our submission.

Sincerely,

Corresponding Auther:

Junichi Hata

Phone: +81-03-3819-1211 (extension; 410), Email: j-hata@tmu.ac.jp, 

Graduate School of Human Health Sciences, Tokyo Metropolitan University, 

7-2-10 Higashi-Ogu, Arakawa-ku, Tokyo, Japan 116-8551

First Auther:

Naoya Hayashi

Department of Radiological Sciences, Graduate School of Human Health Sciences, Tokyo Metropolitan University

7-2-10 Higashi-Ogu, Arakawa-ku, Tokyo, Japan 116-8551

Tel: 03-3819-1211

E-mail: n.hayashi731@gmail.com

---

## [Editor Report · Decision Letter 2]

2 Jan 2024

Identification of the reporter gene combination that shows high contrast for cellular level MRI

PONE-D-23-27098R2

Dear Dr. Hata,

We’re pleased to inform you that your manuscript has been judged scientifically suitable for publication and will be formally accepted for publication once it meets all outstanding technical requirements.

Kind regards,

Kazunori Nagasaka

Academic Editor

PLOS ONE

Additional Editor Comments (optional):

Dear Authors,

Congratulation. We think your manuscript is acceptable for publication in PLoS One.

Again, thank you very much for your contribution to our journal.

We look forward to your future manuscript.

Sincerely,

Plos One
---

## [Editor Report · Acceptance letter]

24 Jan 2024

PONE-D-23-27098R2 

PLOS ONE

Dear Dr. Hata, 

I'm pleased to inform you that your manuscript has been deemed suitable for publication in PLOS ONE. Congratulations! Your manuscript is now being handed over to our production team.

Kind regards, 

on behalf of

Professor Kazunori Nagasaka 

Academic Editor

PLOS ONE